# A Differential Evolution Approach to Optimize Weights of Dynamic Time Warping for Multi-Sensor Based Gesture Recognition

**DOI:** 10.3390/s19051007

**Published:** 2019-02-27

**Authors:** James Rwigema, Hyo-Rim Choi, TaeYong Kim

**Affiliations:** Department of Advanced Imaging Science, Chung-Ang University, Heukseok-dong, Dongjak-gu, Seoul 156-756, Korea; jamesrwigema1@gmail.com (J.R.); funappear@nate.com (H.-R.C.)

**Keywords:** differential evolution, inertial sensors, kinect sensors, dynamic time warping, gesture recognition, heterogeneous sensor data

## Abstract

In this research, we present a differential evolution approach to optimize the weights of dynamic time warping for multi-sensory based gesture recognition. Mainly, we aimed to develop a robust gesture recognition method that can be used in various environments. Both a wearable inertial sensor and a depth camera (Kinect Sensor) were used as heterogeneous sensors to verify and collect the data. The proposed approach was used for the calculation of optimal weight values and different characteristic features of heterogeneous sensor data, while having different effects during gesture recognition. In this research, we studied 27 different actions to analyze the data. As finding the optimal value of the data from numerous sensors became more complex, a differential evolution approach was used during the fusion and optimization of the data. To verify the performance accuracy of the presented method in this study, a University of Texas at Dallas Multimodal Human Action Datasets (UTD-MHAD) from previous research was used. However, the average recognition rates presented by previous research using respective methods were still low, due to the complexity in the calculation of the optimal values of the acquired data from sensors, as well as the installation environment. Our contribution was based on a method that enabled us to adjust the number of depth cameras and combine this data with inertial sensors (multi-sensors in this study). We applied a differential evolution approach to calculate the optimal values of the added weights. The proposed method achieved an accuracy 10% higher than the previous research results using the same database, indicating a much improved accuracy rate of motion recognition.

## 1. Introduction

Human activity recognition using sensor technologies in the computing environments has become an important emerging field of research in recent years of computer application. Countless studies have been conducted on how human activities can be recognized using sensor technologies in computing environments. Recently, the research interest has set focus on a natural user interface, mainly in most human action recognition where vision sensors—such as point grey bumblebee XB3 and Camcube—were used for the recognition process. In our research, we used the combination of a depth camera (Kinect Sensor) and a wearable sensor (inertial sensor), which are capable of capturing human motion in 3D. Analyzing the motion obtained from the sensor to determine user intent is an important process of this type of interface. A typical motion recognition system consists of four major stages: motion (gesture) capture, motion expression, classification, and application, as shown in Figure 1. The sequence data obtained from the user is classified by the pattern recognition technology and then used as inputs of the user’s operation, hence replacing the role of the keyboard and mouse.

To bring the user’s movements into the virtual computer world, many studies using color sensors have been done [1,2]. The main challenges during these studies were to determine object shape, object texture, background, lightning conditions, the distance between the object and the sensor, and changes in viewpoints [3,4]; depth data was proposed as a way to overcome the limitations of color sensors. Also, a time-of-flight (TOF) sensor was introduced, which calculates the distance by measuring the reflected time due to the speed of light [5,6]. Depth sensor data-based research has improved performance in many areas compared to color sensor-based studies but has not been actively used in many areas due to its high price. The Kinect depth sensor was then introduced and widely used during motion recognition methods, enabling a much higher recognition rate than the color sensor-based methods [7]. However, as the demand for the target operation increases gradually, the limitation of these sensors occurs during the recognition of complex movements due to blind spot and occlusions. This is due to the fact that the recognition method of these sensors is fixed in a certain position. To solve this problem, methods using a plurality of sensors or wearable sensors have been studied [8,9]. When increasing the number of sensors or fusing various kinds of sensors, both the number and size of the feature date increases. Many methods have been published to develop algorithms that can be applied to various situations [1,10], and studies on the combinations of sensors suitable for recognizing motion in various situations are still in progress.

Many types of research have been conducted in various fields to read a user’s movement and use the data in the interface. Pattern recognition algorithms are used to extract meaningful information from data and are used in various fields, such as computer vision, speech processing, and motion recognition. Methods used include matching-based dynamic time warping (DTW) [11], the hidden Markov model based on probability modeling [12], conditional random fields (CRFs) [13], and the convolutional neural network (CNN) [14]. The latter is able to effectively learn two-dimensional images and exploit the discriminative features of any gesture. CNN helps us to learn suitable “dynamic” features from skeleton sequences without training millions of parameters afresh, which is especially valuable when there is insufficient annotated training data including the mapping of joint distribution, spectrum coding of joint trajectories, spectrum coding of body parts, and joint velocity weighted saturation and brightness for motion recognition. However, the major problem was how to calculate the optimal values of the weights while the joints were capturing for recognition [15,16,17].

To accurately recognize possible complex movements, a weighted DTW (Dynamic Time Warping) based on multiple sensors was suggested [18]. The proposed method is based on the weighted data and focuses on improving classification accuracy by adjusting weights. Data from multiple sensors and the heterogeneous sensor were used in the configuration. Although many Kinect sensors have high accuracy, there are some limitations during their installations for various sensing environments. Therefore, to overcome these challenges, we used a wearable sensor in addition to Kinect sensors. The major challenge was to discover the appropriate mathematical methods that could be applied in setting the optimal weights. We applied the differential evolution method to calculate the optimal weight value of the feature data extracted for the motion recognition to obtain efficient and accurate results, which could then be applied to various environments.

The differential evolution approach is known to be one of the most powerful reliable optimization algorithms that can be employed to calculate the weight optimal value; this was used for setting the weights of the distance metrics used in a combination to cluster the time series [15].

In [19], a differential evolution approach using an outstanding algorithm was applied to calculate the marginal likelihood of the Gaussian process.

## 2. Dynamic Time Warping 

DTW is a template-matching algorithm used to find the best match for a test pattern out of the reference patterns, where the patterns are represented as a time sequence of features.

### 2.1. Dynamic Time Warping for Gesture Recognition

Let ***R*** = {*r*_1_, *r*_2_... *r_N_*}, *N* ∈ ℕ, and ***T*** = {*t*_1_, *t*_2_, ..., *t_M_*}, *M* ∈ ℕ be a reference and test sequences (sequence of the set of joint positions in our case), respectively. The objective is to align the two sequences in time via a nonlinear mapping. Such a warping path can be illustrated as an ordered set of points, as given below:$$p=\left(p_{1},p_{2},\ldots \ldots \ldots p_{l}\right),\;p_{l}=\left(n_{l},m_{l}\right)$$
where $D_{p}$ is the total cost of the path *p* and $d\;\left(r_{i},\;t_{j}\right)$ measures the distance between elements $r_{i}$ and $t_{j}$. For gesture recognition, the distance can be chosen as the distance between the corresponding joint positions (3D points) of the reference gesture, ***R***, and the test gesture ***T***.

Hence, the optimal path denoted by $p^{*}$ is the path with the minimum total cost. The DTW distance between two sequences is defined by the distance associated with a total cost *D* given in Equation (1) using the optimal path
(1)$$D_{p}=\overset{L}{\underset{l=1}{\sum }}d\;\left(r_{i},\;t_{j}\right),\;$$
where $D_{p}$ is the total cost of the path *p* and $d\;\left(r_{i},\;t_{j}\right)$ measures the distance between elements $r_{i}$ and $t_{j}$. For gesture recognition, the distance can be chosen as the distance between the corresponding joint positions (3D points) of the reference gesture, ***R***, and the test gesture ***T***.

Hence, the optimal path denoted by $p^{*}$ is the path with the minimum total cost. The DTW distance between two sequences is defined by the distance associated with a total cost *D* given in Equation (1) using the optimal path 

(2)$$DTW\left(R,T\right)=D_{p^{*}}\left(R,T\right)$$

The calculation of the optimum path *D*, in consideration of the local path limitation, is as follows:(3)D(i,j)=d(i.j)+min[D(i−1,j−1),D(i−1,j),D(i,j−1)]

The main calculation cost takes place during the calculation process of the optimum adjustment of Equation (3), and although some limitations and dynamic programming could alleviate such issues, the limitation cannot accurately find the results if an optimal result exists outside of the selected data.

### 2.2. Weighted DTW for Multiple Sensors

A weighted distance in the cost computation based on how relevant a body joint is to a specific gesture class was proposed [20]. To incorporate these weights into the cost, the distance function *d*($r_{i},t_{j}$) becomes a weighted average of joints distances between two consecutive frames obtained from the Kinect sensors and the inertial sensors (***T***) and reference frames (***R***)
(4)$$d_{w}\left(r_{i},t_{j}\right)=\;\sum d^{j}\left(r_{i},t_{j}\right)w_{j}^{g}$$
which gives the distance between the *i*th skeleton frame of reference gesture ***R*** and the *j*th skeleton frame of test gesture ***T***, where ***R*** is a sequence known to be in gesture class *g* and ***T*** is an unknown test sequence. 

The relevancy is defined as the contribution of a joint to the motion pattern of that gesture class. To understand a joint’s contribution to a gesture class, we compute its total displacement (i.e., contribution) during the performance of that gesture by a trained user as follows,
(5)$$C_{j}^{g}=\overset{N}{\underset{n}{\sum }}d^{j}\;\left(f_{n-1}^{g},\;f_{n}^{g}\right)$$
where *g* is the gesture index, *j* is the joint index, *n* is the skeleton frame number, and $d^{j}$() computes the displacement of the *jth* joint’s coordinates in feature vectors $f_{n-1}^{g}$ and $f_{n}^{g}$. By summing up these consecutive displacements, one can find the total displacement of a joint in a selected reference action.

Using the total displacement to assess the contribution of a joint in performing a gesture, the weights of action class *g* are calculated using
(6)$$w_{j}^{g}=\;\frac{1-e^{C_{j}^{g}}}{\sum_{k}\left(1-e^{C_{j}^{g}}\right)}$$
where $w_{j}^{g}$ is joint *j*’s weight value for each gesture class *g*. We used the exponential function in order to minimize the loss of gesture displacement, and it iteratively fits a weak gesture to improve the current estimate at each iteration of the gesture.

As a first step of implementing a motion recognition method capable of recognizing various motions, weights were applied to multiple features extracted from the sensors’ data. For the motion of each joint, DTW was used to calculate the similarities of every operation, and once the results had similar behaviors [17], after capturing the user’s information sequences from the sensors, normalization and weighting methods were applied to mitigate the difference between the sequences as shown in Figure 2.

Prior to setting the weights, matching-based DTWs were affected by size variations, which required a normalization based on the overall size and the position of the data. Therefore, the width and the height of the movement was measured by dividing the size of all data by the largest value and the average position.

In the proposed method, we aim to improve gesture recognition accuracy. This is done by assigning higher weighted values to the data with only a lower margin of error, by considering the movements of the joints (directions) and the location of the camera. The size of the dimension of the weight determines the advantage of each element, and it is determined by the dimension of the data used. Naturally, since standard DTW computes the distance of all points with equal penalization of each point regardless of the phase difference, the performance of the gesture recognition system can be improved by providing a weight considering the characteristics of the elements compared to the distance expression. Therefore, a high weight value was assigned to the data with a small error and less noise among the data joint sequences. The weighted values were assigned by calculating the sine value of the two vectors [18].

The sensors used during the research are the Kinect sensors and the inertial sensors (wearable sensors). Kinect is a low-cost real-time depth camera (sensor), capable of projecting a pattern of infrared ray points; it captures the image using the infrared camera and is correlated with a pattern for a known distance. Inertial sensors enable position, orientation, acceleration, and speed of a moving body, determined very precisely in just a single component [21].

Although the method proposed effectively recognizes complex 3D motion, installing a large number of Kinect sensors is a challenge. In addition, when the number of Kinect sensors is reduced, the blind spot occurs during the shooting of gestures due to an insufficient number of Kinect sensors. To overcome this problem without reducing the number of Kinect sensors in use, we opted for using a few sensors that capture data such as angular velocity, acceleration, and magnetic force; these data are then compared with the point of the joints obtained from the Kinect sensors. After acquiring data using the two types of sensors, these data required a normalization process: we applied the differential evolution method for proper weight distribution on the acquired data as follows.

The number of normalized weights = {the number of sample joints (14) + acquired data from wearable devices (3) + the number of sample joints after 1st differential evolution (14)} × the number of frames (*N*). The number of frames varies wildly, sometimes even reaching 100 frames. For example, the number generated frames *(N*) = 40 and the number of normalized weights = (14 + 3 + 14) × 40 = 1240 have to be regularized to get the optimum values. Therefore, as the number of frames increases, it makes it very difficult to find the optimal set of weights.

## 3. Differential Evolution to Optimize the Weights of DTW

In this Section, we propose a differential evolution approach used in the weighted DTW framework, which helped us during the recognition of complex motion. Our research aimed to build robust features using the extracted data obtained from the Kinect and inertial sensors which were used in acquiring the data from the moving joints, and then comparing them to motions within the public database for recognition, as shown in Figure 3.

Differential evolution (DE) is known to be one of the most powerful reliable stochastic real-parameter evolutionary algorithms; it has been used in several applications to solve several arising optimization problems [19,20,22,23,24]. We used the DE technique to calculate the optimal values of weights on weighted DTW for multi-sensors. During the calculation of the optimal weight for each sensor, the fusion of heterogeneous sensor data became too difficult to be analyzed, which is why we applied the differential evolution method. Our method used multiple evaluation criteria to repeatedly select multiple candidates (targeted joints). Our method made it possible to search for the target vector within a large space with multiple candidates, and may be discontinuous once the proper candidate is retrieved. This made it possible to solve the problems presented in [25,26]. In the differential evolution method, we randomly generated pre-existent feature vectors, which were mutated, crossed, and selected, as shown in Figure 4. 

In the differential evolution method, the parameters (weights) to be optimized were initialized and filled with random values to form feature vectors called agents, and as the number of weights increased, they were combined together through a mutation process and formed multiple agents which became the candidates. Candidates groups within the zone of the motion were created and when combined with the new weights of the initial vectors in the zone, they formed a new agent (feature vector) through a crossover; this was done repeatedly while selecting the optimal weights until an appropriate result was achieved. The weight vector required for motion recognition using heterogeneous sensors was determined by a target motion (joints). Each element had a weighted value between 0 and 1, and the set weights determined the importance of the joints in the frame. 

The optimal weight value was obtained by applying the differential evolution represented as parameter *G* in Equation (6):(7)$$w_{j,G}^{g}=\;\frac{1-e^{-C_{j,G}^{g}}}{\sum_{k}\left(1-e^{-C_{j,G}^{g}}\right)}$$
*G* is generated for the currently generated element, the *n*th target feature vector of the target gesture becomes
(8)$$T_{n,G}=\;\left[(x_{1,n},\;G),\;(x_{2,n},G),\;(x_{3,n},G),\ldots \ldots \ldots \ldots \ldots \left(x_{S,n},G\right)\right],$$
where *S* is the features, and *x* is the number of frames. For each parameter, the range values of the parameters must be limited for a finite amount of time. Initial weights of each agent are generated randomly, and similarly, the initial agent is generated and undergoes the mutation process. The parent vector is called the target vector while the vector through the mutation process is the donor vector. By combining the target and the donor vectors, they form a trial vector [22]. To generate the donor vector of the *n*th target feature vector, variable vectors $X_{T_{1}^{n}},$
$X_{T_{2}^{n}},\;X_{T_{3}^{n}\;}$ which are mutually individuals (vectors) are selected randomly; at this time, the selected vectors should not be duplicated. The donor vector $V_{n,G}$ is calculated as follows;
(9)$$V_{n,G}=X_{T_{1,G}^{n}\;}+\alpha \left(\;X_{T_{2,G}^{n}\;}-\;X_{T_{3,G}^{n}\;}\right);$$
where *α* is a mutation factor or the differentiation constant and it is one of the control parameters of DE. In addition, *α* is randomly chosen from [0,1]. The selection process is performed through Equation (10).
(10)$$X_{n,G+1}=\;\{\begin{array}{c} V_{n,G}\;if\;\left(rand_{i,j}\left[0,1\right]<c_{r}\right)\\ X_{n,G}\;\;otherwise \end{array}$$
where *n* = 1,2,3…*G*, and $c_{r}$ is the crossover constant, which is another control parameter. The control parameters of DE are determined by the algorithm designer [19].

In our experiment, the number of parents used was 50 per action, and the best 5 parents were selected. The remaining parents are all initialized as random elements (parent) for the target vector. If a large number was repeated or the result of α function did not change, the reloading was considered complete and the repetition was stopped.

## 4. Experimental Results

For any gesture recognition approach, there is a need for either a private or a public database for reference gesture actions. Among all available public databases, The University of Texas at Dallas Multimodal Human Action Dataset (UTD-MHAD) database of joint actions was used. Gesture recognition experiments, such as accumulating the user’s joint trajectory using convolution neural networks, the distance between the joints of user’s expression using convolution neural networks, the cumulative recording of user’s movements by the use of an image using convolution neural networks, and human action recognition using a depth camera and a wearable inertial sensor using data fusion process of multi-sensors, were studied using this database [15,16,17,27]. UTD-MHAD [27] was considered to be suitable for verification. The inertial sensor was attached to the arm or leg along the main moving body. Table 1 presents the 27 actions which were used for a variation of the motion.

Where from action 1 to 21, the inertial sensor was attached to the right wrist of the subject, and from motion 22 to 27, and it was attached to the right thigh of the subject, as shown in (Figure 5).

Figure 5 indicates the multimodality data corresponding to the action of basketball shoot of the color image, the depth image, the skeleton joint frames and the inertial sensor data. For motion recognition performance can be improved effectively by applying the data fusion of different sensor data which is possible through weighting method though it is difficult to set appropriate weights. We performed weight optimization using a differential evolution method. As a result of the motion recognition experiment using the heterogeneous sensor, two Kinect sensors (Microsoft for Xbox 360, Microsoft corporation model, U.S patent Nos. 6,483,918 and 6,775,708, China) and two wearable inertial sensors (MYOD5, Thalmic labs, Ottawa, Canada) were placed on the wrist and on the thigh, and the UTD-MHAD composed of 27 movements was used. During our experiment, we used 14 samples of the actions within the database, and we applied the differential evolution of approach to the target feature joints, in order to optimize the applied weights during the joint movement actions where more than 1240 weights were to be normalized. We used the joint position sequence extracted from the Kinect of UTD-MHAD and the inertial sensor, which included angular velocity, acceleration, and magnetometer sequences. All motion data (Kinect and inertial sensor) were manually extracted from the beginning to the end of each action. 

### 4.1. Experimental Simulation for Bowling Action

Figure 6 below shows the gesture recognition experiments tested on a bowling game. Through this experiment, the UTD-MHAD was applied to accumulate the user’s joint trajectory while he/she was wearing a wearable sensor (inertia) on the wrist and thigh.

The proposed method is implemented using Matlab (Chung-Ang University, Seoul, South Korea). Firstly, the features were used in a differential evolution method of joint data obtained from the Kinect SDK and the inertial sensor which capture 3 axes (3-axis acceleration, 3-axis angular velocity, and 3-axis magnetic strength); these were measured without considering the change over time. The method of difference evolution determines the usage proportion of each sensor data; thus, the weight value range corresponding to each feature was set between 0 and 1, and the value of the parameter in Equation (10) was set to 0.1 and 1. A total of 27 actions in the UTD-MHAD were used for each operation, and 50 initial parent vectors were used. The experimental results are shown in Figure 7.

Figure 6 shows the bowling behavior of the UTD-MHAD, while Figure 7 shows how the weight of the bowling movement were set using the differential evolution method against the increase of frames. The weights filled with random values changed over the generated frames and converged as shown in Figure 7. Finally, as the number of iterations (frames) increased, an average recognition rate also increased. Therefore, after a total number of 1489 iteration, an average accuracy recognition rate of 99.40% was obtained.

### 4.2. Experimental Results of the Adjusted Weights by Differential Evolution (α = 0.1)

In our experiment, we used all 27 actions (gestures) within the UTD-MHAD. In the experimental results shown in Figure 8a,b, the first 13 actions and the average were considered in (a), while the remaining actions (14) and average were considered in (b). After applying the differential evolution extracted features and setting the value of *α* to 0.1, the small value of *α* limited us and we used only the two vectors’ gaps. This affected the number of generated frames with respect to time. As weights changed slowly, the recognition rate also increased slowly. The number of frames increased while repeating the iterations, and after 178 generations, the average recognition rate scored 83.88%.

### 4.3. Experimental Results of the Adjusted Weights by Differential Evolution (α = 1)

In our experimental results, after the change of the value of α to 1, we were able to use the whole gap of the vector randomly which enables the change of weights rapidly. Figure 9a,b, shows that, after changing the value from α to 1, the same number of generations as that of α = 0.1 were generated in almost the same small period of time (2^−3^) of the time as that which was used when α was 0.1, giving an average accuracy rate of 57%.

In our experiments, we also extended the dimension and further increased the number of frames and used all 27 actions within the database; these were subdivided into two parts, as presented in the two experimental figures, respectively Figure 10a,b. This experiment was conducted when the value of *α* was set to 1, and the results indicated that, after using the same period of time as the results in Figure 8a,b, the number of weights increased rapidly, and the generation increased 8 times compared with those where α was set to 0.1 (from 179 to 1471 generations). Hence, we were able to obtain average accuracy results of 99.4%. In addition, compared to the results obtained during our experiment in Figure 9a,b and Figure 10a,b, it indicated that as the weight value increased, the number of frames being generated also increased, reflecting the same amount of time during the action.

In the study using the UTD-MHAD, according to the previous researches, joint-trajectory map-based using the CNN method obtained results of 89.81% [17]. A CNN based on joint distance map of 88.10% was achieved [16]. The motion history image based on the CNN method was 84% [15]. A depth motion map based on multiple sensors 79.10% was archived [27]. However, the average recognition rates presented by the previous researches were still low. After realizing that the major cause of these low results was due to the complexity in calculating the optimal values of the acquired data from the sensors during the optimization process (as well as the environment of installation of some sensors which provides redundant data due to the blind spot of too many Kinect sensors), we proposed a method which enabled us to reduce the number of Kinect sensors installation and to use inertial sensors instead. We were thus able to overcome the blind spots, and after acquiring data from our sensors, we applied the differential evolution method, which enabled us to calculate the optimal values of the added weights. Our proposed method achieved 99.40%, which indicates a much improved accuracy rate of motion recognition compared to other results as shown in the Table 2. 

## 5. Conclusions/Recommendations

In this paper, we propose a differential evolution method to optimize the weights of DTW and compare it with other motion recognition methods. Since multiple Kinect sensors were constrained to the application environment during their installation, we encountered blind spot challenges and the problem of complex calculations of the optimal weight values of acquired data. During our experiment, we used two Kinect sensors and two wearable inertial sensors (one of each on the wrist and the other two on the thigh) for motion capturing. We used UTD-MHADs for our experiments, and the results of our proposed method can be seen in Figure 7, Figure 8, Figure 9 and Figure 10a,b. A differential evolution method was used to calculate the optimal weights of the acquired data. In our experimental results, an increase of 10% in recognition accuracy was achieved compared to the highest accuracy rate achieved by the previous researcher using the same database. However, we observed a tradeoff in the processing time in order to obtain better results. We would recommend for the future works to consider how optimal time should be minimized without affecting the experimental results (recognition accuracy).

## Figures and Tables

**Figure 1 sensors-19-01007-f001:**
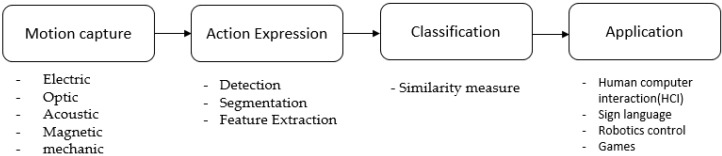
A flow of motion recognition system.

**Figure 2 sensors-19-01007-f002:**
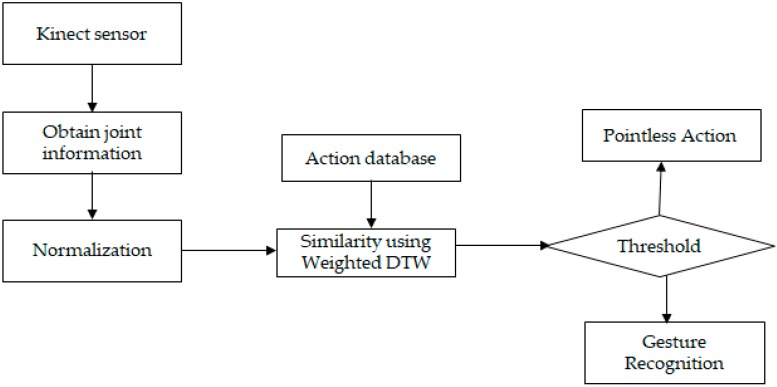
Gesture Recognition with Weighted DTW.

**Figure 3 sensors-19-01007-f003:**
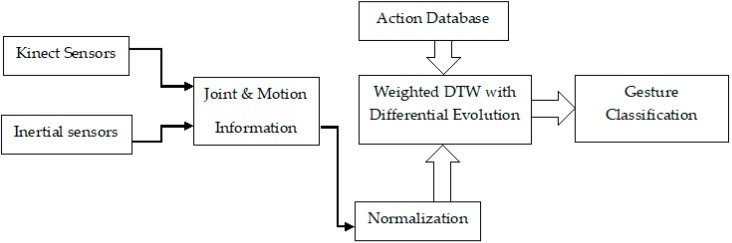
Gesture Recognition of Weighted DTW with Differential Evolution.

**Figure 4 sensors-19-01007-f004:**
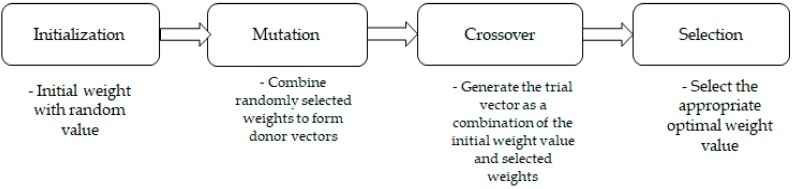
The flow of the differential evolution method.

**Figure 5 sensors-19-01007-f005:**
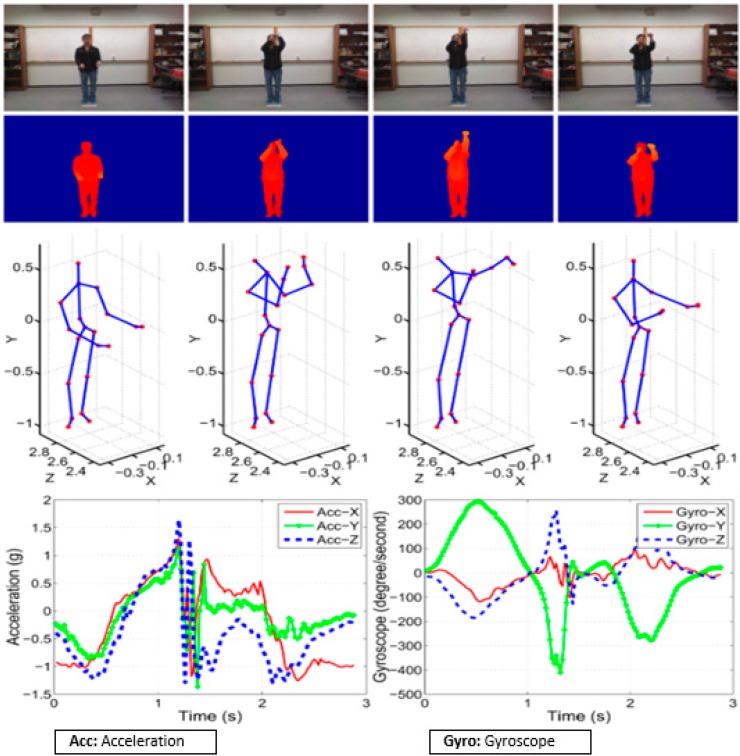
Data configuration of UTD-MHAD [27].

**Figure 6 sensors-19-01007-f006:**
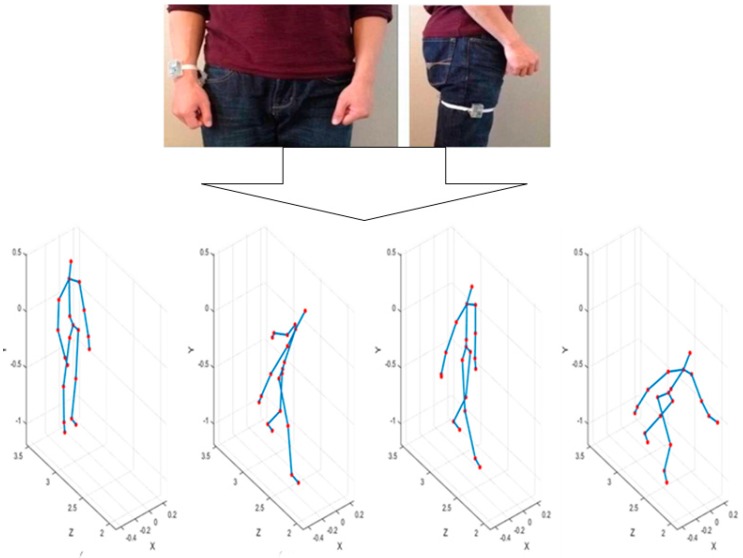
Example of bowling action of UTD-MHAD.

**Figure 7 sensors-19-01007-f007:**
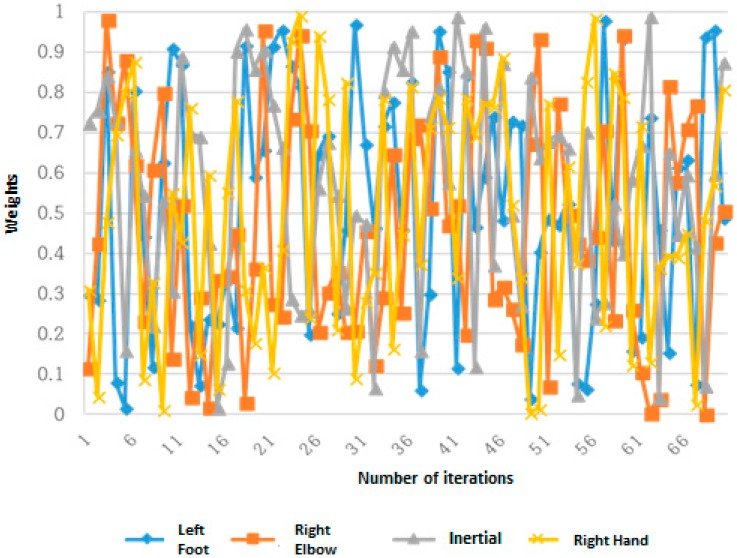
Weights of bowling motion set through the differential evolution method. Experimental Results for Bowling Action with Adjusted Weights by Differential Evolution Where (0.1≤α≤1).

**Figure 8 sensors-19-01007-f008:**
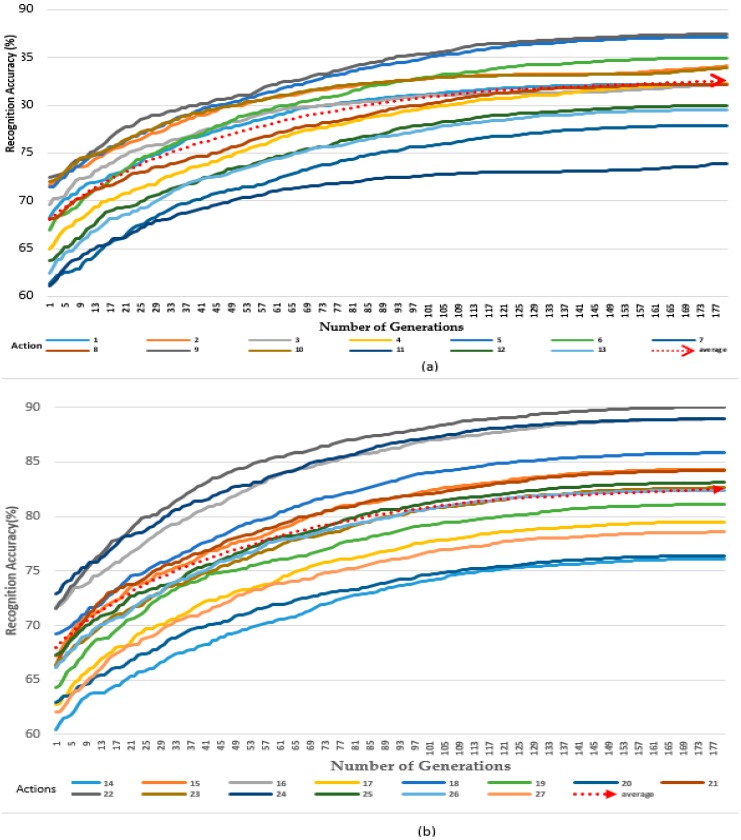
Accuracy with adjusted weights by Differential Evolution (α = 0.1). (**a**) The first 13 actions of the datasets (1 to 13), (**b**) the last 14 actions of the datasets (14 to 27).

**Figure 9 sensors-19-01007-f009:**
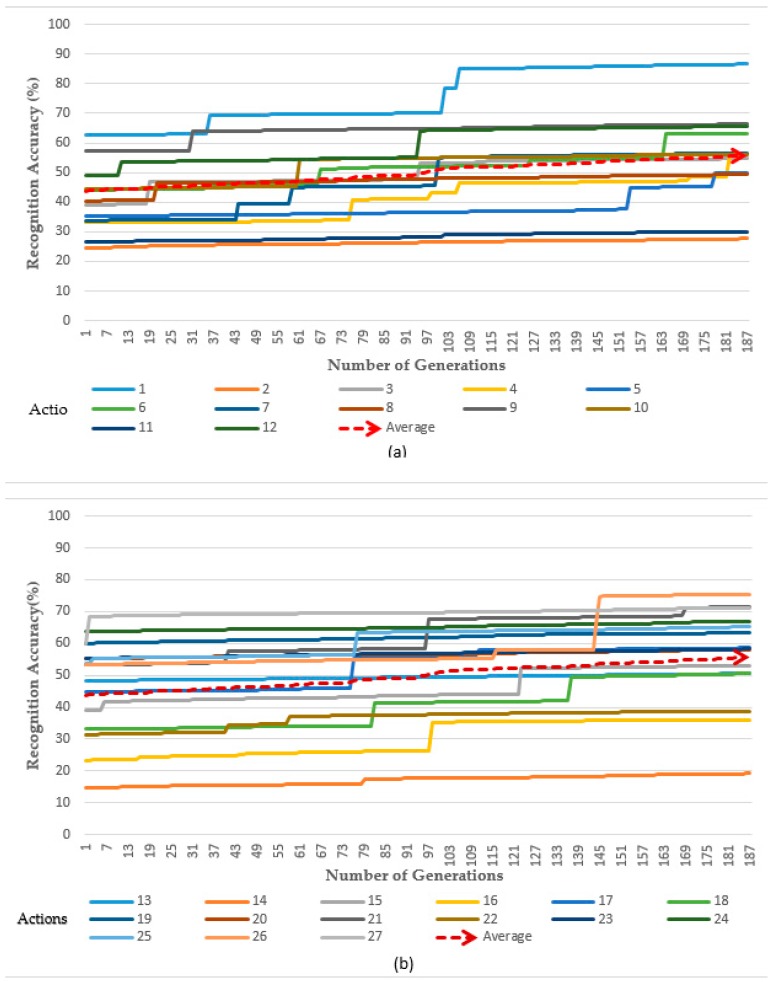
Accuracy with adjusted weights by Differential Evolution (α = 1). (**a**) The first 13 actions of the datasets (1 to 13), (**b**) the last 14 actions of the datasets (14 to 27).

**Figure 10 sensors-19-01007-f010:**
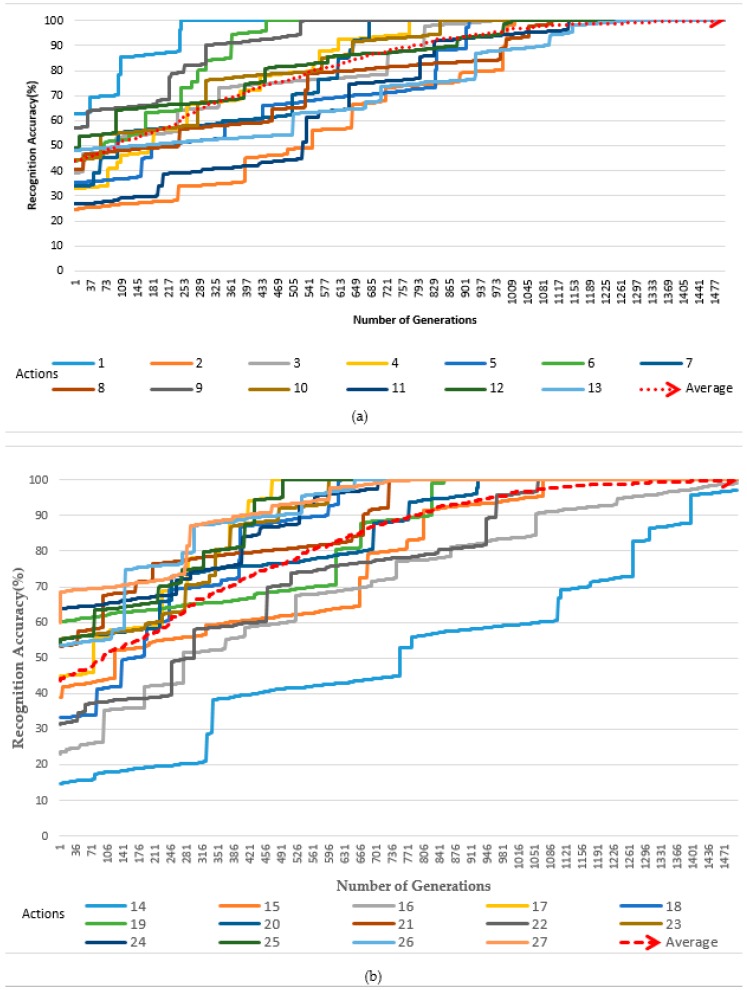
Accuracy with adjusted weights by Differential Evolution (α = 1). (**a**) The first 13 actions of the datasets (1 to 13), (**b**) the last 14 actions of the datasets (14 to 27).

**Table 1 sensors-19-01007-t001:** 27 actions of University of Texas at Dallas Multimodal Human Action Datasets (UTD-MHAD).

S/N	Action
1	Right arm swipe to the left
2	Right arm swipe to the right
3	Right hand wave
4	Two hand front clap
5	Right arm throw
6	Cross arms in the chest
7	Basketball shoot
8	Right hand draw x
9	Right hand draw circle (clockwise)
10	Right hand draw circle (counter clockwise)
11	Draw triangle
12	Bowling (right hand)
13	Front boxing
14	Baseball swing from right
15	Tennis right hand forehand swing
16	Arm curl (two arms)
17	Tennis serve
18	Two hand push
19	Right hand knock on the door
20	Right hand catch an object
21	Right hand pick up and throw
22	Jogging in place
23	Walking in place
24	Sit to stand
25	Stand to sit
26	Forward lunge (left foot forward)
27	Squat (two arms stretched out)

**Table 2 sensors-19-01007-t002:** Experimental results on UTD-MHAD.

Technic Used	Accuracy of Recognition	Characteristics
Joint Trajectory Map	89.81%	Accumulating the user’s joints trajectory, Based on CNN [17].
Joint Distance Maps	88.1%	The distance between the joints of the user’s expression, based on CNN [16].
Motion History Map	84.00%	Cumulative recording of the user’s movement by the use of an image, CNN-based [15].
Depth Motion Map	79.10%	Human action recognition using a depth camera and a wearable inertial Sensor. Data fusion from multi-sensors [27].
Proposed Approach	99.40%	Differential evolution to optimize weights. DTW Based.

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
