# Peer review of "A Differential Evolution Approach to Optimize Weights of Dynamic Time Warping for Multi-Sensor Based Gesture Recognition"

_sensors, 2019, doi:10.3390/s19051007_

Round 1
Reviewer 1 Report
Authors present the development of a robust gesture recognition method that can be used in various environments based on the differential evolution approach that optimizes weighted DTW. With this approach, the authors have been able to improve the performance of the new method is better than that of the traditional methods with effective convergence of heterogeneous sensor data.
The novelty of the reported research is the use of differential evolution method for motion recognition to obtain efficient and accurate recognition performance. They used differential Evolution to compute the optimal weight for individual sensor and fusion of heterogeneous sensors to analyse this complex sensor data. Using wearable sensors such as inertial sensors in combination with Kinect sensors to reduce the number of Kinect sensors due to their limitation as well as using differential evolution method to compute optimal weights of the obtained data is the key strength of their solution. The methodology and the experimental design to evaluate the proposed solution are good. But why use only accuracy as evaluation metric?
In the abstract, the authors state that the performance of DTW gesture recognition is higher than that of conventional methods but by how much? This should have been stated in the abstract.
I would also suggest that the experimental section have subsections according to the experimental evaluation conducted to improve readablity
There are quite a number of language issues/typos, few examples are given below:
Line 24 under taken
Line 25 with in the
Line 48, with in the
Where instead of were in line 256.
Lines 54,55 Methods used includes
Line 65 was suggest..
The article should be edited to correct these mistakes or typos.
Author Response
We greatly appreciate your kind review and detailed comments. According to the suggestions, we have revised the manuscript. For your convenience, your comments and questions are printed in normal font and our replies are in bold font with sections from the text in blue.
We apologize if some the comments are not well responded, it is due to time constraint

Reviewer 2 Report
The evaluation results obtained by the method proposed in this paper are very good. However the overall presentation of the method and description of experiment is not satisfying. In my opinion it is impossible to reproduce the experiment (for example – how the training and validation dataset were composed?). Also the conclusion of the paper is not supported by the data.
Detailed comments:
“In recent years of computer application advancements, lots of studies are under taken of how sensor technologies can replace the natural user interfaces” – what does natural user interfaces mean in this context? All NUI uses sensors of some kind (infrared etc.). Do authors by the term sensors mean “wearable sensors”? Please clarify this.
“which are;” here should be a colon, not semicolon.
Define formally using mathematical terms what is “width and the height of the movement”
How is the submitted paper differ from authors earlier work “Combined Dynamic Time Warping with Multiple Sensors for 3D Gesture Recognition“ – please explain it in the introduction.
Which version of Kinect authors uses in their research?
“we opted using a few sensors” – how many and what type of sensors exactly? According to website http://www.utdallas.edu/~kehtar/UTD-MHAD.html there is only one Kinect and one wearable sensor.
Please explain “Our method makes it possible to search for a large space, it may be discontinuous, and can also be used for problems that vary along the path” – in the current form this sentence is incomprehensible.
“In our experiment, the number of parents used were 50 per action, and the best 5 parents were selected” – why such parameters were selected?
“27 action which are shown in the table below” – this sentence is grammatically incorrect.
“For motion recognition performance can be improved effectively” – maybe “The motion recognition performance can be improved effectively”?
“All motion data (Kinect and inertial sensor) were manually extracted from the starting and the end points.” – what were those “end points”?
What are “three axes acquired from the inertial sensor”?
Figure 7 and 8 – what are “generated frames”?
Figure 5 and Figure 6 are exact copy from website http://www.utdallas.edu/~kehtar/UTD-MHAD.html You cannot reproduce images without giving a credits to the authors. Do you have permission for using them?
Also according to that website “For our multimodal human action dataset reported here, only one Kinect camera and one wearable inertial sensor were used”
Authors of submission stated that “Our work proposed the use of wearable inertial sensors to reduce the number of Kinect sensors” – in order to prove, that internal sensors can reduce number of vision – based sensors you have to have a dataset that has at least two vision based sensors. At this moment you propose a genetic algorithm – based method for DTW parameters calculation.
In my opinion the conclusion is not supported by the data.
Such high recognition rate (99.4%) in not trivial dataset is VERY rear. Please show the confusion matrix and discuss which classes where confused with which and why.
I am not a native speaker but the language quality in this paper should be improved. Sometime it is difficult to understand what authors had on their minds.
Author Response
We greatly appreciate your kind review and detailed comments. According to the suggestions, we have revised the manuscript. For your convenience, your comments and questions are printed in normal font and our replies are in bold font with sections from the text in blue.
we apologize if some the comments are not well responded to. it was due to time constraint

Reviewer 3 Report
In this paper, the authors used differential evolution to optimize weights of dynamic time warping in the context of gesture recognition. Although the idea is correct and may be of interest, the manuscript suffers from a number of weaknesses which prevent it from being publishable in its current form:
1) The experimental section is pretty low-quality and must be improved. First, the experiments are not reproducible, which is not acceptable for a scientific paper (also, the description of the method itself must be improved; see e.g. p. 6, fig. 4 - the evolutionary operators are very vague and unclear). The experimental results are not discussed properly, and the authors did not execute any statistical tests (which I would expect in a randomized approach) to see whether their results are statistically important. Finally, the figures are not self-contained, difficult to read and understand. Overall, I find the experimental analysis section really weak and I think it should be substantially re-worked.
2) As mentioned earlier, the authors should discuss their method (and motivation behind choosing differential evolution) in much more detail (it should be possible to re-implement this method based solely on this manuscript).
3) I suggest performing careful proofreading (perhaps by an English native-speaking colleague).
4) The authors failed to contextualize their work within the state of the art of (hyper)parameter optimization (especially in the context of evolutionary approaches). Just to mention a few techniques which may be used for optimizing hyperparameters of various algorithms:
- https://dl.acm.org/citation.cfm?id=3071208
- https://link.springer.com/chapter/10.1007/978-3-642-20282-7_43
- https://ieeexplore.ieee.org/document/8297018
- https://arxiv.org/abs/1807.07362
Such methods should be at least mentioned and briefly discussed in the related-literature section (which should be extended).
5) I suggest to avoid using "DTW" in the title (without unfolding it).
6) It would be interesting to see the estimated time complexity of the proposed method (alongside its execution time).
Author Response
We greatly appreciate your kind review and detailed comments. According to the suggestions, we have revised the manuscript. For your convenience, your comments and questions are printed in normal font and our replies are in bold font with sections from the text in blue.
we apologize if some of the comments are not well responded to. It is due to time constraint

Round 2
Reviewer 2 Report
Authors have addressed my comments and in my opinion paper can be accepted as it is, however I cannot guarantee its language quality.
Author Response
I would like to take this opportunity to thank you for the good advise and guidance and I have put under considerations all the comments you outlined during your experienced review of my article.
look forward to work with you in further researches with in this field.
Thanks
Best Regards
James Rwigema

Reviewer 3 Report
Unfortunately, the authors did not appropriately address my main concerns, therefore I cannot recommend accepting this manuscript for publication.
Author Response

(The authors gave the same response as above.)

Round 3
Reviewer 3 Report
I appreciate seeing that the authors addressed the majority of my concerns.